# Vaccination against Varicella Zoster Virus Infection in Less Developed Regions of Guangdong, China: A Cross-Sectional Serosurveillance Study

**DOI:** 10.3390/vaccines11030494

**Published:** 2023-02-21

**Authors:** Huimin Chen, Chumin Liang, Xiaorong Huang, Qianqian Ruan, Zhaowan Li, Ximing Hu, Lilian Zeng, Huifang Lin, Jialing Li, Xin Xie, Qi Zhu, Tao Liu, Limei Sun, Jiufeng Sun

**Affiliations:** 1Department of Public Health and Preventive Medicine, School of Medicine, Jinan University, Guangzhou 510632, China; 2Guangdong Provincial Institute of Public Health, Guangzhou 510300, China; 3Guangdong Provincial Center for Disease Control and Prevention, Guangzhou 510317, China; 4School of Public Health, Southern Medical University, Guangzhou 510515, China; 5School of Public Health, Sun Yat-sen University, Guangzhou 510080, China; 6School of Mathematics and Computing Science, Guilin University of Electronic Technology, Guilin 541004, China; 7China Greater Bay Area Research Center of Environmental Health, School of Medicine, Jinan University, Guangzhou 510632, China

**Keywords:** varicella, anti-VZV IgG, seroprevalence, undeveloped, vaccination, Guangdong

## Abstract

Vaccination is the key to prevent varicella zoster virus (VZV) infection in children. Voluntary and self-funded strategies have led to variable vaccination rates against VZV in China. For low-income populations, in particular, the effects of VZV vaccination have been insufficiently estimated. Community-based serosurveillance was conducted in two less developed regions, Zhanjiang and Heyuan, of Guangdong, China. Anti-VZV IgG antibodies in serum were detected by ELISA. The vaccination data were derived from the Guangdong Immune Planning Information System. A total of 4221 participants were involved, of which 3377 were from three counties of Zhanjiang and the other 844 were from one county of Heyuan, Guangdong, China. The total VZV IgG seropositivity rate in vaccinated individuals was 34.30% and 42.76%, while it was 89.61% and 91.62% in non-vaccinated populations of Zhanjiang and Heyuan, respectively. The seropositivity rate increased gradually with age, reaching ~90% in the >20- to 30-year-old group. The VarV vaccination rates of children aged 1–14 years were 60.47% for one dose and 6.20% for two doses in Zhanjiang, and 52.24% for one dose and 4.48% for two doses in Heyuan. Compared with the non-vaccinated group (31.19%) and one-dose group (35.47%), the positivity rate of anti-VZV IgG antibodies was significantly higher in the two-dose group (67.86%). Before the VarV policy was reformed, the anti-VZV IgG positivity rate was 27.85% in the one-dose-vaccinated participants, which increased to 30.43% after October 2017. The high seroprevalence in participants was due to infection of VZV in Zhanjiang and Heyuan, not vaccination against VZV. Children aged 0–5 years are still vulnerable to varicella, so a two-dose vaccination program should be implemented to prevent onward transmission of VZV.

## 1. Introduction

Varicella zoster virus (VZV) is a human herpesvirus that belongs to the alpha-herpes virus family. It is a highly infectious pathogen that can be transmitted through air droplets from the respiratory tract or direct contact with varicella blister fluid [1,2]. Varicella (chickenpox) mainly occurs as a primary VZV infection during childhood followed by a variable latent period, after which it may reactivate to cause herpes zoster (shingles) in adults [3]. The primary infection of VZV usually presents as a relatively mild, self-limiting illness with papules, macules, blisters, and scabs. It may also cause pneumonia, encephalitis, conjunctivitis, thrombocytopenic purpura, and other complications in severe cases [4]. Post-herpetic neuralgia is the serious complication of herpes zoster, which is extremely painful and shows no response to treatment; notably, it may lead to VZV encephalitis and vasculitis [5,6].

Vaccination with the varicella attenuated live vaccine (VarV) is the most economical and effective measure to prevent varicella in children [7]. Although VarV has been widely available since 1974 [8], public health policies regarding immunization against varicella are quite variable worldwide. The varicella vaccine has been included as part of national immunization programs in some countries [9,10]. However, VarV is a category II vaccine in China, meaning it is voluntary and self-funded [11]. In Guangdong, China, the Guangdong provincial varicella vaccination program (2017) was rolled out for children aged <14 years [12]. Although the incidence of varicella has significantly decreased in developed countries, where VarV vaccination has been implemented within the last few decades [10,13,14], there are still frequent reports of varicella outbreaks in schools annually [15,16,17]. Seroprevalence surveys in vaccinated and non-vaccinated populations are the key to evaluating the effects of vaccination programs on the prevention and control of varicella transmission in the community. Most of archived seroprevalence data for VZV infection are from high-income countries; reports from low- to middle-income populations are rare [18]. Similarly, in China, most investigations into the prevalence rate of VZV infection have been from first-line cities, e.g., Beijing and Shanghai [19,20]. Zhanjiang and Heyuan are economically less developed areas in Guangdong. In 2018, the vaccination rates of VarV in Zhanjiang and Heyuan were 6.30% and 4.20%, respectively, lower than the average level of 11.53% in Guangdong. Therefore, there is a high risk of varicella outbreaks in susceptible populations in both cities, and understanding the seroprevalence of varicella in these two less developed regions is important.

Serosurveillance and comprehensive analysis of vaccination data may guide policy reform of vaccination programs in less developed regions of China, as well as other similar areas in Southeast Asia.

## 2. Materials and Methods

### 2.1. Study Design

The survey was conducted by stratified cluster random sampling between November 2019 and December 2020. First, three counties from Zhanjiang (Leicheng, Wushi, and Qishui) and one county from Heyuan (Heping) were selected as sampling work fields using proportional to the population size (PPS), then the communities were sampled by PPS, after which residents of all age groups were selected as monitoring subjects through simple random sampling. The Guangdong provincial varicella vaccination program (2017) was suitable for children aged <14 years. The first dose should be given at 1–2 years old and the second dose at 4–6 years old. Those who did not complete the two doses at the specified ages should complete a course through emergency vaccination and the inoculation interval should be ≥3 months. The inclusion criteria for subjects were as follows: (1) a local resident for at least 3 months; (2) with good physical health; (3) for children aged 1–14 years, the brand of VarV was BCHT (BCHT, Changchun, China). The exclusion criteria for subjects were as follows: (1) refused collection of venous blood; (2) had unclear varicella vaccination information (missing date and dose of vaccination); (3) immunocompromised patients and recipients of blood and blood products during the last 6 months. A total of 4226 citizens were recruited. Five citizens were excluded as they refused to donate venous blood. Among the children aged 1–14 years old, 52 had incomplete vaccination information and were excluded from further analysis.

### 2.2. Information Collection

The basic information of the participants was collected by questionnaire survey, including age, gender, residential address, and ID number. If the participants were children, the questionnaire was filled out by their guardians. The entire varicella vaccination history of each participant aged 1–14 years was taken from Guangdong Immune Planning Information System (http://10.79.0.253/nipisgd/login.action, accessed from 10 January to 10 February 2022) in Guangdong CDC, including the date and the doses of vaccination. The sampling and vaccination history were matched with the personal ID number.

### 2.3. Serum Sampling

Peripheral venous blood (5 mL) was taken from the participants in a local hospital. For newborns, 5 mL umbilical cord blood was collected. All blood samples were initially centrifuged at 3000 rpm for 5 min. The serum was transferred to a new sterilization tube marked with the sample name and date of blood collection. All serum samples were frozen at –20 °C until testing.

### 2.4. Varicella Zoster Virus IgG Antibody (Anti-VZV IgG) Test

Enzyme-linked immunosorbent assay (ELISA) was performed on all samples using the SERION ELISA classic Varicella zoster virus IgG antibody detection kit (Virion/Serion, Wurzburg, Germany) to test the presence of anti-VZV IgG. The experimental procedure was conducted strictly according to the kits’ manual instructions. The standard serum with anti-VZV IgG in PBS was the positive control and the negative control was PBS without anti-VZV IgG. The serum samples were initially diluted at a ratio of 1:100 and then 100 µL of dilution was added to the determination plate for 1 h incubation at 37 °C. After incubation, the plate was washed with an automatic plate washing machine (BioTek 405 TS, Winooski, VT, USA) using lotion buffer solution, repeated four times. This was followed by 100 µL IgG enzyme-labeled antibodies being added to each well and then incubated at 37 °C for 30 min. After one-cycle washing step, 100 µL of each substrate was added to each well and incubated again at 37 °C for 30 min. Finally, 100 µL of termination solution was added. All the ELISA results were determined using an enzyme-labeled instrument (BioTek Epoch, Winooski, VT, USA), and the data were presented as the OD (optical density) value at the wavelength λ = 405 nm. The cutoff value refers to the manual instructions. An OD value above the cutoff value was considered positive. The OD value for children aged 1–14 years with anti-VZV IgG positive was converted to the antibody concentration values (mIU/mL) using software from SERIO. The test range of the kit was 50–2000 mIU/mL. The results greater than 2000 mIU/mL were labeled as 2000 mIU/mL.

### 2.5. Statistical Analysis

All data information was introduced into combined Excel files and analyzed by SPSS version 20.0 software (SPSS Inc., Chicago, IL, USA). The anti-VZV IgG positivity rate in different cities, age groups, and vaccination histories was compared by Chi-square test. The geometric antibody titers (GMCs) of different groups were compared by one-way ANOVA, and multiple comparisons were performed using the LSD test. *p* < 0.05 was considered statistically significant.

## 3. Results

### 3.1. Demographic Information of Participants

A total of 4221 participants were enrolled in the study. Among them, 3377 were from Zhanjiang (Leicheng, 910; Wushi, 1328; and Qishui, 1139), including 1263 men (37.40%) and 2114 women (62.60%). The remaining 844 were from Heping, Heyuan, including 366 men (43.36%) and 478 women (56.64%) (Appendix A). In addition, 269 neonatal umbilical cord blood samples (6.37%) were collected in the two cities.

### 3.2. Seroprevalence of Anti-VZV IgG in Non-VarV-Vaccinated Participants

To understand the epidemiology characteristics of varicella in Zhanjiang and Heyuan, all participants were divided into two groups: vaccinated and non-vaccinated. A total of 3786 (89.69%) participants had no vaccination history for VarV from Zhanjiang and Heyuan. The serum anti-VZV IgG positivity rates of participants across all age groups were 89.61% (11.63–96.80%) and 91.62% (22.00–97.75%) in Zhanjiang and Heyuan (χ^2^ = 0.224, *p* = 0.621), respectively (Table 1). Vertical transmission of anti-VZV IgG was detected in newborns (89.59%). The seropositivity rates in the three age groups (0–5 y, 5–10 y, and 10–20 y) were lower than in the other age groups (Figure 1). Of particular concern is that the seropositivity rate in children aged 0–5 years was the lowest and it increased gradually with age, reaching ~90% in the >20–30 years old group. The differences in the seropositivity rate among age groups were statistically significant in Zhanjiang (χ^2^ = 656.9, *p* < 0.001) and Heyuan (χ^2^ = 263.0, *p* < 0.001), whereas no significant difference within the same age groups was found between the two cities (Table 1).

### 3.3. Seroprevalence of Anti-VZV IgG in VarV-Vaccinated Participants

In this study, 258 and 268 children aged 1–14 years had complete vaccination information in Zhanjiang and Heyuan, respectively. In Zhanjiang, the varicella vaccination rate was 60.47% for one dose and 6.20% for two doses. In Heyuan, the varicella vaccination rate was 52.24% for one dose and 4.48% for two doses. The one-dose VarV vaccination rate of children aged 1–2 years was 75.00% in Zhanjiang and 54.17% in Heyuan. The two-dose VarV vaccination rate of children aged 4–6 years was 10.37% in Zhanjiang and 6.21% in Heyuan (Table 2).

The positivity rate of anti-VZV IgG increased with the doses of varicella vaccination. The seropositivity of anti-VZV IgG was statistically significant in both cities (Zhanjiang, χ^2^ = 9.896, *p* = 0.007; Heyuan, χ^2^ = 6.514, *p* = 0.038) (Appendix A). No significant difference was found between the non-VarV group and the one-dose group (Zhanjiang, χ^2^ = 1.809, *p* = 0.179; Heyuan, χ^2^ = 0.230, *p* = 0.632). Compared with the two-dose group, the positivity rate of anti-VZV IgG in the one-dose group was also significantly higher (Zhanjiang, χ^2^ = 6.224, *p* = 0.013; Heyuan, χ^2^ = 5.532, *p* = 0.019) (Figure 2).

### 3.4. Persistence of Anti-VZV IgG in Vaccinated Participants

The participants aged 1–14 years who were anti-VZV IgG positive were divided into three groups: non-vaccinated (zero dose), one dose, and two doses. There were 63 participants with no vaccinations, 105 participants with one dose of vaccination, and only 19 participants with two doses of vaccination (Appendix A).

We determined the dynamic tendency of anti-VZV IgG titers in vaccinated participants (Figure 3B,C). According to the different time intervals between sampling and vaccination, the 105 participants with one dose of vaccination were divided into four groups: ≤1 y, >1–3 y, >3–5 y, and >5 y. The 19 participants with two doses of vaccination were only divided into three groups: ≤1 y, >1–3 y, and >3–5 y. Among the participants of the one-dose group, the anti-VZV IgG level was not significantly different at different time intervals (F = 0.311, *p* = 0.818). However, the anti-VZV IgG level of the 63 non-VarV participants was higher than that of the one-dose group (t = 2.241, *p* = 0.026) (Figure 3A,B). No significant difference was found between the non-VarV and two-dose-vaccinated groups (t = 1.154, *p* = 0.252) (Figure 3A,C).

Then, the non-vaccinated and one-dose participants were both divided into three age groups: 1–3 y, 4–6 y, and 7–14 y. There were no significant differences in the 1–3 y, 4–6 y, and 7–14 y groups (non-vaccinated, F = 0.047, *p* = 0.954; one dose, F = 1.376, *p* = 0.257) (Figure 3D,E).

### 3.5. Effects of Varicella Vaccination Policy Reform

In October 2017, the Guangdong CDC officially issued the Guangdong provincial varicella vaccination program for children (2017). Before 2017, Guangdong provincial public authorities suggested that children over 1-year old receive one dose of varicella vaccination [21]. In this study, we found the anti-VZV IgG positivity rate was 27.85% among the 158 children with one dose of vaccination before 2017, and none had accepted two doses of vaccination. After 2017, anti-VZV IgG positivity rate reached 30.43% among the 138 children with one dose of vaccination and 67.86% among the 28 children with two doses of vaccination (Appendix A). No significant difference was found between the seropositivity of anti-VZV IgG of one dose of vaccination before and after the promulgation of the two-dose vaccination program (χ^2^ < 0.001, *p* = 0.625).

## 4. Discussion

In 2014, the estimated nationwide vaccination rate of varicella in children was only 20.84% in China [22]. In Guangdong, the total vaccination rate of VarV in school-age children was even lower, at 11.53% in 2018. In addition, the vaccination rate of VarV was variable due to the regional socioeconomic level [23]; for example, the vaccination rate of VarV in the Pearl River Delta region, which is a more developed region of Guangdong, was 22.82% in 2018, whereas it was only ~10% in other regions with a low Gross Domestic Product (GDP) (the GDP of eastern, western, and north Guangdong was 26.50%, 28.80%, and 35.30% of the Pearl River Delta, respectively). Zhanjiang and Heyuan are located in western and northern Guangdong, respectively. According to the annual report of the Statistical Yearbook in Guangdong (2020), the GDP per capita of Zhanjiang and Heyuan were 44.00% and 50.30% of the GDP per capita in Guangdong, and 53.90% and 61.70% of that in China; therefore, both cities belong to economically less developed areas. According to the surveys, outbreaks of varicella in Guangdong mainly occurred in the Pearl River Delta region [24,25,26], and only a few cases were reported in geographically remote areas of Guangdong. The epidemiological trends of varicella in Guangdong are still unclear and do not reflect the effects of VarV vaccination. A few studies on anti-VZV antibody surveillance have been conducted in the Pearl River Delta region of Guangdong [27,28,29], but no vaccination data were interpreted in these studies.

This study was conducted to understand the status of VZV infection and to assess the effects of anti-VZV vaccination in Zhanjiang and Heyuan. First, we found that the seropositivity rate of anti-VZV IgG in newborns reached 85%, which was obtained through vertical transmission from mothers. These data are similar to those from a study in Korea [10], in which the seropositivity rate was 75% after birth but declined rapidly after 4–6 months and reached ~10% 12 months later. A study in China also found that anti-VZV IgG could be detected in cord blood, but in few infants 6 months after delivery [30]. These data indicated that the anti-VZV IgG obtained through mother-to-child transmission only lasted ~12 months. Second, we found that the positivity rate of anti-VZV IgG was lower in children under 5 years in both the one-dose-vaccinated and non-vaccinated populations, which probably meant a lower protection effect for one dose of vaccination, and no significant difference from the non-vaccinated population, in this age group. However, the presence of anti-VZV IgG increased rapidly in teenagers and adults, reaching ~90% after >20–30 years. This was consistent with a cross-sectional investigation in China [31]. Regarding the lower vaccination rate in both cities, we believe this was due to natural infection with VZV, not persistent antibodies from VarV vaccination. Third, the total rates of anti-VZV IgG in Zhanjiang and Heyuan were 89.6% and 91.6%, respectively. These were consistent with studies using ELISA assays from Kunming (85.80%) and Beijing (84.50%) [19,32]. Anti-VZV IgG levels in South Korea (93.90%) [10] and Israel (90.20%) [33] were comparable with those in China, but higher than in Singapore (52.90%) [34], Iran (78.50%) [35], and India (68.22%) [36]. Although there may have been bias in these comparisons of total rates of anti-VZV IgG levels against varicella with other countries due to the age structure of the study participants in those countries, the methodology, and the vaccination policy and vaccination coverage rates for certain ages, there was a trend of varicella seroprevalence. A further comparison study focusing on VarV vaccine or viruses could be circulated in different geographical areas.

We found that the one-dose VarV vaccination rates in participants aged 1–14 years from Zhanjiang and Heyuan were 60.47% and 52.24%, respectively, which are comparable to those in Suzhou (66.79%) and Hangzhou (70.06%) [37,38]. However, our results showed that the two-dose VarV vaccination rate was low in Zhanjiang (6.20%) and Heyuan (4.48%). In both cities, the seropositivity rate of anti-VZV IgG between the non-vaccinated group and the one-dose group showed no difference, while it was lower than that in the two-dose group. The same was found in previous studies [34,37,38]. Moreover, a low VarV coverage rate can easily lead to epidemic outbreaks of varicella, and archived studies have shown that varicella outbreaks often occur in areas with a high coverage of one-dose vaccination [17,39]. Thus, the effect of a single dose of VarV is limited, and a second vaccination of VarV is required to enhance the protective effect of vaccination and prevent onward transmission of VZV.

In this study, we found that the anti-VZV IgG level of non-vaccinated children was higher than that of the one-dose children. We believed that this may be because non-vaccinated children may experience a natural infection of VZV during varicella outbreaks in their kindergartens and primary schools. We also determined the persistence of anti-VZV IgG induced by VarV vaccination. The results showed that the anti-VZV IgG persisted for 5 years in children after one dose of varicella vaccination. However, a report from Jiangsu province showed that the titers of varicella antibody decreased gradually after vaccination [40]. Cohort studies from the USA and Japan also showed a long-term persistence of antibodies in healthy children after one dose of varicella vaccine [41,42]. Others reported that individuals vaccinated more than 30 years ago showed no evidence of waning immunity, while some have reported that people become vulnerable in as few as 6 years [43]. Primary vaccination failure is common and frequently reported worldwide. A study from the USA showed that the rate of primary vaccination failure was 24% [44]. A high rate of primary vaccine failure is likely to play a key role in breakthrough infection cases. We compared the anti-VZV IgG level of different age groups (1–3 y, 4–6 y, and 7–14 y) in non-vaccinated and one-dose children, and we found that the anti-VZV IgG level in children aged 4–6 years was lower than in the 1–3 years and 7–14 years groups. A study by Yuyang Xu et al. also showed that the GMCs for VZV IgG in 4- to 6-year-old children were lower than those in children younger than 4 years and aged 7–14 years [38], which indicated that children aged 4–6 years should receive a second dose of VarV in a timely manner.

Reform of the vaccine immune program may impact the dynamics of population vaccination. In 2006, the Advisory Committee on Immunization Practices (ACIP) recommended a routine two-dose varicella immunization schedule, and it was employed first in the USA [45,46,47]. Many countries have now started a two-dose varicella vaccination program, including the USA, Germany, and Greece [48]. At present, most cities in China still implement a one-dose VarV program. An archived study showed that one-dose VarV is not enough to prevent outbreaks of varicella [49]. A two-dose VarV procedure has been introduced in a few cities, e.g., Beijing in 2012 and Hangzhou in 2014 [38]. In Guangdong, two doses of VarV were recommended for children aged 1–14 years until 2017 [20]. From the results of this study, there was no difference between the seropositivity rate before and after promulgation of the procedure, while it increased among those who completed two doses of vaccination. This was consistent with a study showing that the incidence of varicella in children has reduced since the varicella vaccination program was adjusted from one dose to two doses [50].

There were some limitations to this study. First, breakthrough infection of VZV in vaccinated participants was not assessed due to the limited number of participants in Guangdong. Second, the influence of factors that may affect the vaccination of VarV was not included in the analysis in this study, e.g., income and education level. Third, this study was a term cross-sectional study, which can describe the characteristics of varicella immunization of different groups but cannot draw a conclusion on the dynamics of both varicella and VarV vaccination. Finally, we only sampled the serum of participants from Zhanjiang and Heyuan. Geographical differences should be considered in further studies.

## 5. Conclusions

In this study, we determined the population seroprevalence of varicella in healthy people from Zhanjiang and Heyuan, and we found a high anti-VZV IgG positivity rate in teenagers and adults. However, the positivity rate of anti-VZV IgG was low in children aged 0–5 years. A comparison assay between vaccinated and non-vaccinated populations indicated that the high seropositivity was due to natural infection of VZV in teenagers and adults, not persistent anti-VZV IgG induced by vaccination. Nevertheless, two doses of varicella vaccine are essential for children in low-income regions. A comprehensive analysis incorporating varicella incidence data will be essential to assess the protection effects of VarV vaccination in the real world in future.

## Figures and Tables

**Figure 1 vaccines-11-00494-f001:**
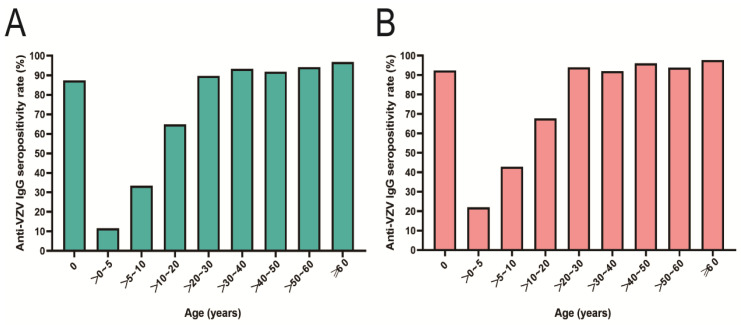
(**A**) Seroprevalence of anti-VZV IgG in non-VarV-vaccinated participants in Zhanjiang; (**B**) seroprevalence of anti-VZV IgG in non-VarV-vaccinated participants in Heyuan.

**Figure 2 vaccines-11-00494-f002:**
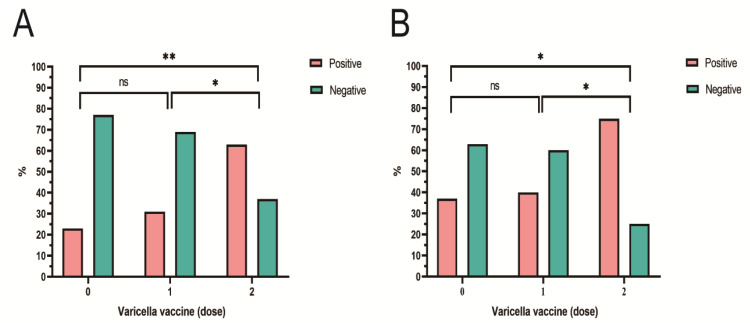
(**A**) Seroprevalence of anti-VZV IgG in VarV-vaccinated participants in Zhanjiang; (**B**) seroprevalence of anti-VZV IgG in VarV-vaccinated participants in Heyuan. %, the proportion of the participants with anti-VZV IgG; 0, non-vaccinated group; 1, one-dose-vaccinated group; 2, two-dose-vaccinated group; ns, *p* > 0.05; * *p* < 0.05; ** *p* < 0.01.

**Figure 3 vaccines-11-00494-f003:**
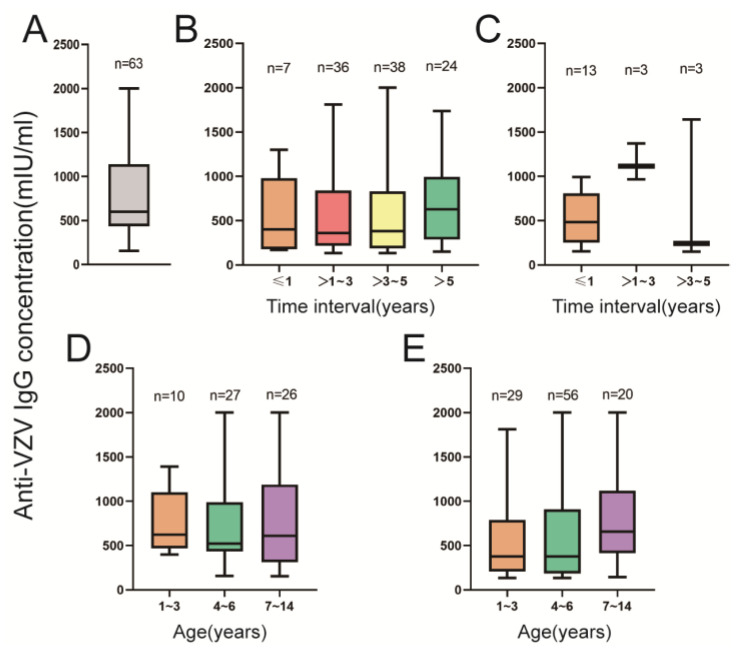
The persistence of anti-VZV IgG in participants with different vaccination histories. (**A**) The anti-VZV IgG level of non-vaccinated participants; (**B**) the anti-VZV IgG level of one-dose participants; (**C**) the anti-VZV IgG level of two-dose participants; (**D**) the anti-VZV IgG level of non-vaccinated participants in different age groups; (**E**) the anti-VZV IgG level of one-dose-vaccinated participants in different age groups.

**Table 1 vaccines-11-00494-t001:** Seroprevalence of anti-VZV IgG in non-VarV-vaccinated participants.

Age (Years), No. (%)	Zhanjiang (*n* = 3194)	Heyuan (*n* = 692)	*p* Value
0	131 (87.33)	110 (92.44)	0.173
>0–5	5 (11.63)	11 (22.00)	0.186
>5–10	11 (33.33)	30 (42.86)	0.357
>10–20	59 (64.84)	21 (67.74)	0.769
>20–30	417 (89.68)	31 (93.94)	0.626
>30–40	332 (93.26)	93 (92.08)	0.682
>40–50	112 (91.80)	97 (96.04)	0.194
>50–60	176 (94.12)	92 (93.88)	0.935
≥60	1692 (96.80)	87 (97.75)	0.847
Total	2935 (91.89) 2862 * (89.61)	572 (82.66) 634 * (91.62)	0.621

* This number represents the age-adjusted overall anti-VZV IgG seropositivity.

**Table 2 vaccines-11-00494-t002:** The coverage of the varicella vaccine for participants aged 1–14 years in both cities.

Age (Years) No. (%)	Zhanjiang				Heyuan			
	S	N	1	2	S	N	1	2
1–2	4	3 (75.00)	3 (75.00)	-	24	13 (54.17)	13 (54.17)	-
3	62	31 (50.00)	31 (50.00)	-	44	29 (65.91)	29 (65.91)	-
4–6	135	104 (77.04)	90 (66.67)	14 (10.37)	161	93 (57.76)	83 (51.55)	10 (6.21)
7–14	57	34 (59.65)	32 (56.14)	2 (3.51)	39	17 (43.59)	15 (38.46)	2 (5.13)
Total	258	172 (66.67)	156 (60.47)	16 (6.20)	268	152 (56.72)	140 (52.24)	12 (4.48)

S, the number of participants in this age group; N, the number of participants vaccinated; 1, one-dose-vaccinated group; 2, two-dose-vaccinated group.

## Data Availability

Not applicable.

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
