# Peer review of "Vaccination against Varicella Zoster Virus Infection in Less Developed Regions of Guangdong, China: A Cross-Sectional Serosurveillance Study"

_vaccines, 2023, doi:10.3390/vaccines11030494_

Round 1

Reviewer 1 Report

Manuscript ID Vaccines-2091658

Review Report

Title: Vaccination against varicella zoster virus infection in less developed regions of Guangdong, China: a cross-section serosurveillance study

This study is focusing on the serosurveillance of Varicella (Chicken pox) in two counties Zhanjiang and Heyuan of Guangdong, China. ELISA test is performed using IgG antibody detection kit from total of 4221 participants serum sample. The results shows that the seropositivity is higher among the >20-30 years age group. I recommend the manuscript for publication with minor revision.

1. The introduction is very clear only recommendation is to add some literature about previously used ELISA test using VZV IgG antibody.

2. Author should specify the positive and negative control used in the ELISA test.

Author Response

Response to Reviewer 1 Comments

Point 1: The introduction is very clear only recommendation is to add some literature about previously used ELISA test using VZV IgG antibody.

Response 1: Thanks for your suggestions. Detection of VZV IgG antibody by using ELISA were common used in various studies, we list some literature in methods section.

Point 2: Author should specify the positive and negative control used in the ELISA test.

Response 2: Revised accordingly. The positive and negative control information was added in methods section.  “2.4. Varicella zoster virus IgG antibody (Anti-VZV IgG) test” as “The standard serum with anti-VZV IgG in PBS was the positive control and the negative control was PBS without anti-VZV IgG.”

Reviewer 2 Report

There are several typos, such as "Zhangjiang" in abstract, which might be "zhanjiang", and the website"http://10.79.0.253/nipisgd/login.action" could not be opened.

What does the p-value in table1 stand for? between two regions within each age group? There are too many statistical tests, it is hard to find useful information from the overwhelming information, such as figure 1, just keep important ones, plz explain more on figure 3 in discussion

Author Response

Response to Reviewer 2 Comments

Point 1: There are several typos, such as "Zhangjiang" in abstract, which might be "zhanjiang", and the website"http://10.79.0.253/nipisgd/login.action" could not be opened.

Response 1: These typing mistakes were revised accordingly. The website "http://10.79.0.253/nipisgd/login.action" belongs to the internal database, which can only be used by internal users. Here we showed the resources of information.

Point 2: What does the p-value in table1 stand for? between two regions within each age group? There are too many statistical tests, it is hard to find useful information from the overwhelming information, such as figure 1, just keep important ones, plz explain more on figure 3 in discussion

Response 2: The p-value in Table 1 stands for the comparison of seroprevalence of anti-VZV IgG within each age group between two regions. It shows that no significant difference within the same age groups between two regions.

Figure 1 was revised accordingly, we indicated the differences between three age group (0~5y, 5~10y, 10~20y) with other age groups in figure legends.

More explanation about Figure 3 was added, and discussed on conclusion section, see in revised text.

Reviewer 3 Report

I congratulate the authors on a study that should contribute to the creation of a varicella vaccination policy in China. However, the manuscript require major revision. My suggestions are listed bellow:

 Introduction

1.     The introduction is generally too long, it needs to be shortened. I suggest shortening the text (lines 52-70 can be moved to discussion).

2.     Lines 55-6. Many countries still implement universal vaccination of children against varicella, not only USA and Korea. I suggest you don't cite which countries implemented universal vaccination, just  add some appropriate references.

3.     Lines 72-80. The aim of the study is too general, it should be clearly and concisely stated with one sentence.

4.     Methods:

5.     Study design: Minimum age of participants was not listed.

6.     Why didn't you combine and analyze the data for both provinces (Zhanjiang and Heian) into one study cohort?

7.     Exclusion criteria „unclear varicella vaccination information“- please explain to readers what you mean?

8.     Please explain why you did not exclude samples from immunocompromised patients and recipients of blood and blood products during the last 6 months? These are common exclusion criteria for serosurveys.

9.     Lines 98-102 should be moved to Introduction.

10. Lines 116-129- please include appropriate references for the lab. tests used. In that case you may shorten this paragraph.

Results:

1.     Lines 139-42 should be moved to Methods. I suggest to start from line142.

2.     Line 167. What does an incomplete vaccination history mean if all the data about vaccination history of participants, as you claim, is taken automatically from Guandong Immune Planing Information System? Did you collect data about vaccinations from the study participants directly ( or asked them to provide documents about vaccinations)?

3.     Number of excluded study participants should be reported in Methods, not in Results (lines 167-168).

4.     Line 169. Please avoid summarizing vaccination rates ( 66.67%), instead, just report vaccination rate for I and II dose of vaccine.

5.     Table 2. Why putting headings of the column in the legends? Please fix this.

6.     Fig 3 ( panels a, b,c). I am afraid that the time interval between sampling and previous vaccination was not indicated and therefore this results are difficult to analyse. I would avoid summarizing, i.e. showing all (unvaccinated, those who received one and those who received two doses)- Fig 3a, instead I suggest showing three panels – unvaccinated (a), those with one dose (b) and those with two doses (c), but for the vaccinated it is necessary to indicate the time intervals between dates of sampling and receiving vaccines. And why combining and analyzing the data for both provinces (Zhanjiang and Heian) into one study cohort here and not in earlier analysis (Results, lines 137-189)

·        If you revise your results accorrding to previous suggestions, this would certainly add weight to the conclusions you stated in lines 210-219.

Discussion is missing!

1.     Line 220 – I suppose you should replace Conclusions with Discussion

2.     Lines  221-232 , I suggest to move ( merge) this paragraph with similar text in the Introduction section. I would start with the listing main results starting from the sentence in the line 234 and to elaborate/discuss the results in the following paragraphs.

3.     Line 249-250. I suggest you rephrase this sentence: not all children and not all elderly are immunocompromised!!! They are only prone to serious clinical forms of varicella and its complications.

4. Lines 250-254. I suggest avoiding comparisons of collective immunity ( total rates of anti VZV IgG levels) against varicella with other countries without specifying the age structure of study participants in those countries, study methodology as well as the vaccination policy and the vaccination coverage rates in certain ages.

5.     Line 265. Hard to tell (see comments no 6 for Results)

6.     Lines 296-304, Limitations should be moved before conclusions  ( lines 287-295).

Author Response

Response to Reviewer 3 Comments

Introduction

Point 1: The introduction is generally too long, it needs to be shortened. I suggest shortening the text (lines 52-70 can be moved to discussion).

Response 1: Revised accordingly.

Point 2: Lines 55-6. Many countries still implement universal vaccination of children against varicella, not only USA and Korea. I suggest you don't cite which countries implemented universal vaccination, just  add some appropriate references.

Response 2: This statement was revised as “Varicella vaccine has been included as part of national immunization program in some countries”.

Point 3: Lines 72-80. The aim of the study is too general, it should be clearly and concisely stated with one sentence.

Response 3: The sentence was revised as “The serosurveillance and comprehensive analysis with the vaccination data may guide the policy reforming of vaccination program in less developed regions of China, as well as rest of similar areas in Southeast Asia.”

 Methods

Point 1: Study design: Minimum age of participants was not listed.

Response 1: Supplemented accordingly.

Point 2: Why didn't you combine and analyze the data for both provinces (Zhanjiang and Heian) into one study cohort?

Response 2: In our study, Zhanjiang and Heyuan were two independent cities of Guandong Province. The total sampling population and proportion of age groups were different between them. In addition, the vaccination rate and economic conditions were also different with each other. That is why two cities were analyzed separately.

Point 3: Exclusion criteria „unclear varicella vaccination information“- please explain to readers what you mean?

Response 3: The Exclusion criteria ”unclear varicella vaccination information” means the children who have varicella vaccination but the information including the date or the doses of vaccination are missing. Revised as “(1) refused collection of venous blood. or (2) had an unclear varicella vaccination information (missing date and dose of vaccination).”

Point 4: Please explain why you did not exclude samples from immunocompromised patients and recipients of blood and blood products during the last 6 months? These are common exclusion criteria for serosurveys.

Response 4: We initial think it normal exclusion criteria. Accurately, we implemented it during the sampling. This information was added in revised text.

Point 5: Lines 98-102 should be moved to Introduction.

Response 5: Revised accordingly.

Point 6:  Lines 116-129- please include appropriate references for the lab. tests used. In that case you may shorten this paragraph.

Response 6: Indeed, lab. tests information was added accordingly.

Results

Point 1: Lines 139-42 should be moved to Methods. I suggest to start from line142.

Response 1: Revised accordingly.

Point 2: Line 167. What does an incomplete vaccination history mean if all the data about vaccination history of participants, as you claim, is taken automatically from Guandong Immune Planing Information System? Did you collect data about vaccinations from the study participants directly ( or asked them to provide documents about vaccinations)?

Response 2: An incomplete vaccination history means the children have varicella vaccination but the information including the date or the dose of vaccination is missing. All the vaccination information was extracted from Guangdong Immune Planning Information System with personal ID number. The initial data of vaccination will match with our sampling individuals. The missing date and dose of vaccination was considered as incomplete vaccination history.

Point 3: Number of excluded study participants should be reported in Methods, not in Results (lines 167-168).

Response 3: Revised accordingly.

Point 4: Line 169. Please avoid summarizing vaccination rates (66.67%), instead, just report vaccination rate for I and II dose of vaccine.

Response 4: Revised accordingly. “Among them, varicella vaccination rate were 60.47% for 1-dose, 6.20% for 2-dose and 52.24% for 1-dose, 4.48% for 2-dose, in Zhanjiang and Heyuan, respectively.”

Point 5:  Table 2. Why putting headings of the column in the legends? Please fix this.

Response 5: Revised accordingly.

Point 6: Fig 3 ( panels a, b,c). I am afraid that the time interval between sampling and previous vaccination was not indicated and therefore this results are difficult to analyse. I would avoid summarizing, i.e. showing all (unvaccinated, those who received one and those who received two doses)- Fig 3a, instead I suggest showing three panels – unvaccinated (a), those with one dose (b) and those with two doses (c), but for the vaccinated it is necessary to indicate the time intervals between dates of sampling and receiving vaccines.

And why combining and analyzing the data for both provinces (Zhanjiang and Heian) into one study cohort here and not in earlier analysis (Results, lines 137-189). If you revise your results according to previous suggestions, this would certainly add weight to the conclusions you stated in lines 210-219.

Response 6: Thanks for your suggestions. Figure 3 was revised accordingly. Three new panels were added to indicate the latency of anti-VZV IgG antibodies, unvaccinated, one-dose and two-dose vaccinated groups, and we also showed the time interval between sampling and previous vaccination, see in revised Figure 3.

Here we combined the vaccination history data from both cities based on the age group were much concentrated between 0-14 years old, the comparable proportion in each age group and limited number of individuals. In particular for residents who receive two dose vaccines, there are only 19 residents in total from two cities.

Finally, I would thanks again for your suggestions, and hope the reviewer would agree with us in this case.

Discussion is missing!

Point 1:  Line 220 – I suppose you should replace Conclusions with Discussion

Response 1: Revised accordingly.

Point 2: Lines  221-232 , I suggest to move ( merge) this paragraph with similar text in the Introduction section. I would start with the listing main results starting from the sentence in the line 234 and to elaborate/discuss the results in the following paragraphs.

Response 2: Revised accordingly.

 Point 3:  Line 249-250. I suggest you rephrase this sentence: not all children and not all elderly are immunocompromised!!! They are only prone to serious clinical forms of varicella and its complications.

Response 3: Deleted accordingly.

Point 4: Lines 250-254. I suggest avoiding comparisons of collective immunity (total rates of anti VZV IgG levels) against varicella with other countries without specifying the age structure of study participants in those countries, study methodology as well as the vaccination policy and the vaccination coverage rates in certain ages.

Response 4: Thanks for your suggestions. This part was changed as “Anti-VZV IgG levels in South Korea (93.90%) [10] and Israel (90.20%) [34] were comparable with that in China, while higher than that in Singapore (52.90%) [35], Iran (78.50%) [36] and India (68.22%) [37]. Although there are potential bias among these comparisons of total rates of anti VZV IgG levels against varicella with other countries due to the age structure of study participants in those countries, methodology as well as the vaccination policy and the vaccination coverage rates in certain ages, it showed a trend of varicella seroprevalence, which may demand a further comparison study either focus on VarV vaccine or viruses circulated in different geographical areas.”

 Point 5: Line 265. Hard to tell (see comments no 6 for Results)

Response 5: Revised accordingly.

 Point 6:  Lines 296-304, Limitations should be moved before conclusions  ( lines 287-295).

Response 6: Revised accordingly.

Round 2

Reviewer 3 Report

The authors revised the manuscript according to the suggestions. I believe that  the quality of the revised manuscript has been improved enough to be accepted for publication. However, more extensive editing of the English language and style is needed in order to improve the readability and comprehensibility of the manuscript.

Author Response

Thanks for your suggestions. We have improved  the English language and style in our manuscript.